# Molecular Mechanism of Xylogenesis in Moso Bamboo (*Phyllostachys edulis*) Shoots during Cold Storage

**DOI:** 10.3390/polym11010038

**Published:** 2018-12-27

**Authors:** Changtao Li, Lingling Xuan, Yuming He, Jie Wang, Hui Zhang, Yeqing Ying, Aimin Wu, Antony Bacic, Wei Zeng, Lili Song

**Affiliations:** 1Sino-Australia Plant Cell Wall Research Centre, The State Key Laboratory of Subtropical Silviculture, School of Forestry and Biotechnology, Zhejiang A&F University, Lin’an, 311300, China; 648677@163.com (C.L.); 18268823164@163.com (L.X.); ym-he@hotmail.com (Y.H.); jiewang524@163.com (J.W.); 15058108752@163.com (H.Z.); yeqing@zafu.edu.cn (Y.Y.); abacic@unimelb.edu.au (A.B.); 2Guangdong Key Laboratory for Innovative Development and Utilization of Forest Plant Germplasm, College of Forestry, South China Agricultural University, Guangzhou 510642, China; wuaimin@scau.edu.cn; 3ARC Center of Excellence in Plant Cell Walls, School of BioSciences, the University of Melbourne, Parkville VIC 3010, Australia; 4La Trobe Institute of Food and Agriculture, La Trobe University, Bundoora, VIC 3083, Australia

**Keywords:** Moso bamboo shoots, cold storage, secondary cell wall, xylogenesis, comparative transcriptomic analysis

## Abstract

A bamboo shoot is the immature stem of the woody grass and a nutritious and popular vegetable in East Asia. However, it undergoes a rapid xylogenesis process right after harvest, even being stored in a cold chamber. To investigate the molecular regulation mechanisms of xylogenesis in Moso bamboo (*Phyllostachys edulis*) shoots (MBSes) during cold storage, the measurement of cell wall polymers (cellulose, hemicellulose, and lignin) and related enzyme activities (phenylalanine ammonia lyase (PAL), cinnamyl alcohol dehydrogenase (CAD), peroxidase (POD), and xylan xylosyltransferase (XylT)) and transcriptomic analysis were performed during cold storage. It was noticed that cellulose and lignin contents increased, while hemicellulose content exhibited a downward trend. PAL, CAD, and POD activity presented an upward trend generally in MBS when stored at 4 °C for 16 days. XylT activity showed a descending trend during the stages of storage, but slightly increased during the 8th to 12th days after harvest at 4 °C. Transcriptomic analysis identified 72, 28, 44, and 31 functional unigenes encoding lignin, cellulose, xylan biosynthesis enzymes, and transcription factors (TFs), respectively. Many of these secondary cell wall (SCW)-related genes showed higher expression levels in the later period of cold storage. Quantitative RT-PCR analysis of the selected genes conformed to the expression pattern. Our study provides a comprehensive analysis of MBS secondary wall biosynthesis at the molecular level during the cold storage process. The results give insight into the xylogenesis process of this economically important vegetable and shed light on solving this problem of the post-harvest industry.

## 1. Introduction

Moso bamboo (*Phyllostachys edulis*) shoots (MBSes) are immature, expanding culms emerging from nodes of the (pseudo-)rhizome of the bamboo plant, mainly in regions of Southern and Eastern China. Because of abundant dietary fiber and distinctive flavor, MBSes have been valued as a popular vegetable and commercially important crop species throughout Asian history. There are approximately 1400 identified bamboo species around the world, and most of them produce shoots [1]. However, as rapidly growing subterraneous immature stems, MBSes deteriorate rapidly, which results in xylogenesis and then decays, resulting in quality loss after harvest. This process occurs even when they are stored at low temperature, as evidenced by lignin deposition within the vascular bundles [2]. Luo et al. [3] also reported lignin and cellulose accumulation in fresh bamboo shoots during cold storage, accompanied by increases in phenylalanine ammonia lyase (PAL), cinnamyl alcohol dehydrogenase (CAD), and peroxidase (POD) activity.

Lignin is a complex polymer formed mainly by three phenylpropanoid monolignols (*p*-coumaryl alcohol, coniferyl alcohol, and sinapyl alcohol) [4]. The monolignol biosynthesis pathway starts with a PAL reaction that synthesizes cinnamate derivatives and accomplishes a POD reaction that forms the units of the lignin polymer [3,5,6]. Subsequently, the lignin polymer is deposited into the secondary cell wall (SCW) matrix, which also consists of cellulose and hemicellulose biopolymers [7]. It has been reported that most of the biosynthesis genes for cellulose, hemicellulose, and lignin, as well as the regulatory machinery (e.g., transcription factors), are upregulated when SCWs are deposited [8,9]. A study of *Arabidopsis* using reverse genetic analysis of an Irregular Xylem (*IRX*) mutant led to the identification of SCW-specific cellulose synthase (*CesA*) genes (*IRX1*/*CesA8*, *IRX3*/*CesA7*, and *IRX5*/*CesA4*), xylan biosynthetic genes (*IRX7*, *IRX8*, *IRX9*, *IRX10*, *IRX14*, and *IRX15*), and lignin biosynthetic genes (*IRX4* and *IRX12*) [10,11,12,13,14,15,16]. Furthermore, these SCW-related structural genes are regulated under a concert of transcription factor (TF) networks [17,18], including NAC/SND, MYB, and other families of TFs. These SCW biosynthetic genes, as well as TFs, may vary with different species, which makes the whole mechanism more complicated [9].

In fast-growing bamboo shoots, candidate genes involving transcription factors, plant hormone metabolism, cell cycle regulation, and cell wall biosynthesis and degradation have been identified using high-throughput short-read sequencing [19,20]. Gamuyao et al. [21] further characterized the transcripts associated with phenylpropanoid biosynthesis in mature shoot tissues. Peng et al. [22] published the genome of bamboo shoots, provided information on potential candidate genes during the MBS fast-growth process, and speculated on the molecular basis of the physiological process. He et al. [20] and Zhang et al. [23] generated a transcriptome by taking advantage of the annotated Moso bamboo genome sequencing and delivered a molecular basis underlying this phenomenon of sequentially elongated internodes from base to top. However, little is known about the coordinated molecular mechanism of secondary wall thickening and lignification in MBS after harvest.

In this study, we comprehensively investigated the xylogenesis process at the physiological, biochemical, and transcriptomic level of MBSes after harvest during cold storage, and identified functional unigenes encoding secondary wall thickening and xylogenesis. The results may provide a theoretical foundation for solutions to this crop storage process, as well as for solutions for other vegetables in the post-harvest industry [23].

## 2. Materials and Methods

### 2.1. Plant Material, Treatment, and Storage

Moso bamboo (*Phyllostachys edulis*) shoots (freshly excavated culms) were harvested from a plantation in Lin’an, Zhejiang province, China, in March 2016, were transported with ice to the laboratory within 5 h after post-harvest, and were precooled overnight at 8−10 °C. Uniformed shoots (length: ~30 cm, diameter: 4–5 cm) without any blemishes or disease were selected for experimentation. Before treatment, about 3 cm from the cut end of each shoot was removed with a sharp kitchen knife. The shoots, weighing about 2500 g, were subsequently sorted into bundles (~6 shoots per bundle). Each bundle was placed into a plastic container and stored at 4 ± 1 °C for up to 16 days with ~95% relative humidity in the dark. There were three technical replicates and 6 shoots per replicate. During storage, the middle part (about 9 cm) of the MBSes were collected to measure the composition of cellulose, lignin, and hemicellulose, and the activities of PAL, CAD, POD, and xylan xylosyltransferase (XylT). For isolation of RNA, MBS samples, which were obtained from those stored at 4 °C on day 0 (stage 1), day 8 (stage 3), and day 16 (stage 5), were mixed and stored at −80 °C until transcriptome analysis.

### 2.2. Sectioning of Stems

Shoot middle sections from three stages (stage 1, 3, 5) were cut and immediately transferred into fixative buffer (1.6% (*v*/*v*) paraformaldehyde and 0.2% (*w*/*v*) glutaraldehyde in 25 mM sodium phosphate, pH 7.2). Stem sections (50–80 μm) were sliced using a Leica VT1000S vibratome with 3% agarose as support, stained for 1–2 min in 0.02% Toluidine blue O (Sigma-Aldrich, St. Louis, MO, USA), rinsed in distilled water, and then mounted in 50% glycerol.

### 2.3. Measurements of Cellulose, Hemicellulose, and Lignin Contents

The measurement method was adopted from the description of Zhao et al. [24], with some slight modifications for plant cell wall components. MBS samples were dried in the oven and ground into fine powder with a kitchen blender. Dewaxing powders, which were utilized for the following measurement, were extracted with toluene-ethanol (2:1, *v*/*v*) in Soxhlet for 6 h in reflux (92 °C). Dewaxing powders were subjected to sulfuric acid hydrolysis as specified in a standard TappiT222 om-02 for acid-insoluble lignin. The acid soluble lignin could be measured by absorption of ultraviolet radiation [ε205 = 110 L (g cm)^−1^]. The dewaxing powders were delignified in sodium chlorite (pH 4.5, adjusted using acetic acid, 76 °C) for 4 h, leaving holocelluloses (hemicellulose and cellulose). The cellulose content was determined via the method of Kurschner–Hoffner. Dewaxing powders (1 g, dry weight) were hydrolyzed with nitric acid–ethanol repeatedly in boiling water until the fiber whitened and the alcoholic nitric acid solution was discarded, and fresh solution was added after each cycle, from which the aforementioned nitric acid–ethanol solution was obtained by mixing one volume of 60% (*w*/*w*) nitric acid solution with four volumes of 95% absolute ethyl alcohol. After four cycles, the cellulose was washed, dried, and weighed. Finally, the difference between holocelluloses and cellulose contents was defined as the hemicellulose content of the powder.

### 2.4. PAL, CAD, and POD Activity

PAL activity was measured according to the following procedure described by Jiang [25]. Crude enzyme was obtained from about 5 g of frozen tissue powder with 15 mL of 0.1 M borate buffer (pH 8.8) containing 0.5 g of polyvinylpyrrolidone (PVP). The homogenized mixture was centrifuged for 15 min at 14,000× *g*, and the supernatant was collected for the determination of PAL enzyme activity. The reaction mixture containing 0.8 mL supernatant, 2 mL of 0.2 M borate buffer (pH 8.8), and 1 mL 0.02 M l-phenylalanine was incubated for 1 h at 30 °C. Finally, 0.5 mL of 6 M HCl was added to terminate the reaction. One unit was defined as the amount of enzyme that caused a change of 0.1 in OD_290_ per hour per gram at optimal temperature. Three independent replicates were conducted for each treatment.

CAD activity was performed according to the procedure of Luo et al. [3]: 5 g of frozen tissue powder was extracted by 10 mL of 0.1 M phosphate buffer saline (PBS, pH 6.25, containing 15 M β-mercaptoethanol, 2 % polyethylene glycol (PEG), and 0.1 g of PVP). After being centrifuged at 18,000× *g* for 20 min, the supernatant was used for the enzyme activity assay. The assay mixture containing 0.2 mL supernatant and 800 μL reaction mixture (10 mM nicotinamide adenine dinucleotide phosphate (NADP) and 5 M trans-cinnamic acid) was measured at the absorbance at 340 nm. One unit of CAD activity was defined as a change in OD_340_ per hour per gram. For each treatment, three independent replicates were conducted.

POD activity was carried out according to Lurie et al. [26], with a slight modification. About 0.4–0.5 g frozen tissue powder was extracted with 5 mL PBS (0.05 M, pH 7.8). The homogenized mixture was centrifuged at 8,000× *g* for 10 min, and the supernatant liquid was collected as the enzyme activity assay. The assay mixture containing 0.5 mL of extract and 3 mL reaction mixture (100 mM, pH 6.0 PBS, 2-Methoxyphenol, and 30% H_2_O_2_) was measured at the absorbance at 470 nm. One unit of POD activity was defined as a change in OD_470_ per minute per gram. For each treatment, three independent replicates were conducted.

### 2.5. Assay of Xylan XylT Activity

#### 2.5.1. Preparation of Microsomal Membranes

Microsomes were isolated from the middle of MBSes following the procedure of Zeng et al. [27], with minor modifications. In brief, plant tissues were homogenized with extraction buffer (1 mL per gram tissue) containing 50 mM *N*-2-hydroxyethylpiperazine-N-ethane-sulphonicacid-potassium hydroxide (HEPES-KOH) (pH 6.8), 0.4 M sucrose, 1 mM dithiothreitol (DTT), 5 mM MnCl_2_, 5 mM MgCl_2_, and a completely ethylenediaminetetraacetic acid EDTA-free proteinase inhibitor cocktail tablet (Roche, Basel, Switzerland), in a mortar by grinding. The homogenate was filtered through two layers of miracloth (Merck Millipore, Billerica, MA, U.S.), and the filtrate was centrifuged at 3000× *g* for 10 min at 4 °C. The supernatant was centrifuged at 30,000× *g* for 30 min at 4 °C. The 30,000 *g* pellets were resuspended in homogenization buffer and stored at −80 °C.

#### 2.5.2. Anthranilic Acid Labeling of Xyl_5_

The Xyl_5_ assays (Megazyme, Wicklow, Ireland) were labeled at their reducing termini with anthranilic acid (AA) according to the method described by Alwael et al. [28]. Excess derivatization reagent was removed by mixing with diethyl ether (three times), and the AA-labeled Xyl_5_–AA in the bottom layer was purified using a Sep-Pak C18 cartridge (Waters, Milford, MA, U.S.).

#### 2.5.3. Assay of Xylan XylT Activity

The assay of XylT activity with Xyl_5_-AA acceptors was performed in a reaction mixture containing 50 μL microsomal membranes, 1% Triton-100, 0.1 mM cold uridine diphosphate UDP–Xyl, and 0.01 mM Xyl_5_–AA. After incubation at room temperature for about 6 h, the reaction was terminated with 1 μL acetic acid and 1 μL 0.5 M EDTA and centrifuged at 13,040× *g* for 1 min. The supernatant was filtered through a 0.22 μm Ultrafree-MC filter (Merck Millipore, Billerica, MA, U.S.) and centrifuged at 9,060× *g* for 1 min. The products were analyzed by reversed-phase (RP) chromatography on Thermo U3000 Series HPLC systems and a Fluorescence Detector-3000 (Ex_320nm_, Em_420nm_). The Xyl_5_–AA products were separated on a 4.6 mm × 150 mm, 5 μm Aglient Eclipse XDB-C18 RP column at a flow rate of 0.5 mL min^−1^ and a column temperature of 30 °C using a gradient program set to 0 to 2 min (8% B), 2 to 20 min (8% to 20% gradient B). Mobile phase A was 50 mM sodium acetate buffer (pH 4.3), and mobile phase B was acetonitrile.

### 2.6. RNA Extraction, Library Construction, and RNA-Seq Analysis

Total RNA of six samples in MBSes were extracted with a plant total RNA kit (TIANGEN^®^). A NanoDrop^®^ ND-1000 (Thermo Scientific, Waltham, MA, USA) was used to determine the quality and concentration of RNAs, and RNA integrity was further confirmed via electrophoresis on 1% agarose gels. About 30 μg of total RNA from each sample (stage 1, 3, and 5) was used for Illumina sequencing at Biomarker Technologies (Beijing, China). Total mRNA was isolated by an NEB Next Poly (A) mRNA Magnetic Isolation Module (NEB, E7490) or a MICROBExpres Bacterial mRNA Enrichment Kit (Invitrogen, AM1905). All procedures for cDNA library construction were carried out on the NEB Next mRNA Library Prep Master Mix Set for Illumina (NEB, E6110) and the NEB Next Multiplex Oligos for Illumina (NEB, E7500). Sequencing of the purified libraries was performed by an Illumina GA-II (Illumina Inc., Chicago, IL, USA).

### 2.7. Transcriptome Data Analysis

After the RNA was sequenced, trimming adapters obtained high-quality clean reads and got rid of low-quality sequencing data defined as having more than 10% bases with a *Q*-value < 20 and reads with unknown bases. For high-quality reads, they were all aligned to the Moso bamboo reference genome by a Basic Local Alignment Search Tool (BLAST)-like alignment tool (BLAT). For functional annotation, the unigenes were aligned by four public protein databases: Nr (non- redundant protein database in NCBI National Center for Biotechnology Information)), Nt (nonredundant nucleotide database in NCBI), Swiss-Prot, and Kyoto Encyclopedia of Genes and Genomes pathway database (KEGG) (E-value ≤ 1e^−5^). Additionally, based on the Nr annotation, gene ontology (GO) classification was carried out by Blast2GO software (version 2.3.5, http://www.blast2go.de/) with an E-value ≤ 1e^−5^. For differentially expressed genes, reads per kilobase per million reads (RPKM) values were used to calculate gene expression levels [29]. Consequently, statistical comparisons of RPKM values between different samples were conducted using the method described by Audic and Claverie [30]. The DESeq package was used to obtain the “base mean” value for identifying differentially expressed genes (DEGs). False discovery rate (FDR) ≤ 0.01 and the absolute value of log2 ratio ≥ 1 were set as thresholds for detecting significant expression of genes differences [31]. For the analysis of unigenes from MBSes that were related to metabolic pathway genes, unigenes were closely examined according to a search for standard gene names and synonyms in the functional annotations of unigenes. Each search result was verified via BLAST.

### 2.8. Validation of RNA-Seq Data by Quantitative Real-Time RT-PCR (qRT-PCR)

Unigenes related to metabolic pathway synthesis were selected for validation by quantitative real-time PCR (qRT-PCR). Total RNA was extracted for qRT-PCR analysis from stages 1, 3, and 5 according to the procedures described above. The sequences of these selected genes were obtained in the Moso bamboo genome database. Moreover, the primer set for each transcript was obtained by Primer Quest (http://www.ncbi.nlm.nih.gov/tools/primer-blast/). *NTB* was used as the reference gene for all target genes [32], qRT-PCR was carried out by using ChamQ™ SYBR^®^ qPCR Master Mix in a 10 μL volume with CFX96™ Real-Time System (Bio-Rad). Each reaction was conducted as follows: 95 °C for 1 min, 45 cycles of 95 °C for 10 s, 57 °C for 10 s, and 72 °C for 20 s. The melt curve was obtained by heating the amplicon from 65 to 95 °C. All the samples were tested in triplicate, and the experimentation was conducted on three biological replicates to ensure their reproducibility and reliability.

### 2.9. Statistical Analysis

All experimentations followed completely randomized designs. The data were tested by analysis of variance (ANOVA) using SPSS Version 18.0. Least significance differences (LSDs) were calculated to compare significance at the 5% level.

## 3. Results

### 3.1. Changes in Cellulose, Hemicellulose, and Lignin Content of MBSes during Cold Storage

The main components of the cell wall structure in MBSes are cellulose, hemicellulose, and lignin. During the 16-day cold storage process, the cellulose content increased from 24.10% to 28.90%, and lignin increased from 11.85% to 15.62% (Table 1). Interestingly, the hemicellulose content decreased from 35.57% to 28.92%. Moreover, a transverse section taken from the middle segment of MBSes stored at 4 °C on days 0, 8, and 16 (stages 1, 3, and 5, respectively), was observed in vascular bundles during the cold storage time (Figure 1). The vascular bundles showed little toluidine blue staining on day 0, whereas the staining showed deeper color as storage time progressed during cold storage. The evaluation statistics of sequencing data in the three stages of 4 °C can be found in Table 2.

### 3.2. The Enzymatic Activities of PAL, CAD, and POD in MBSes during Cold Storage

As the key enzymes of lignin biosynthesis, PAL, CAD, and POD activities were measured over the storage process. In this study, we found that PAL, CAD, and POD activity presented an upward trend generally in MBSes when stored at 4 °C for 16 days (Figure 2). After 16 days, the activities of PAL, CAD, and POD in MBSes stored at 4 °C increased by 327.2%, 626.1%, and 105.4% compared to the samples before storage.

### 3.3. A Comparison of the Xylan XylT Activity in MBSes during Cold Storage

To investigate changes in xylan XylT activity during the storage process, the middle of MBS tissues from 0, 4, 8, 12, 16 days during storage at 4 °C were collected and analyzed. The XylT activity showed a descending trend during cold storage, but slightly increased in the 8–12 days after being harvested at 4 °C (Figure 3).

### 3.4. Transcriptomic Analysis in MBSes during Cold Storage

To study the lignification of MBSes during three stages of cold storage, three transcriptome libraries of stages 1, 3, and 5 were constructed for high-throughput sequencing, and 41.6 million, 44.0 million, and 47.0 million reads of each stage were generated, respectively (Table 3). To identify the genes corresponding to these clean reads in each library, the reads were mapped to the reference genes expressed in the Moso bamboo genome. Mapping results showed that 36,191,746 (87.15%), 38,481,918 (87.36%), and 41,134,475 (87.55%) reads from each library matched the reference genome (Table 3). Unique matched reads to the genome were 33,702,857 (81.15%), 36,257,017 (82.31%), and 38,784,805 (82.55%) in the three libraries. The alignment statistics of these reads from stages 1, 3, and 5 in MBSes are shown in Table 3. Compared to reference genes, the mapped reads were all above 87%, indicating the relative high quality of these data. The alignment statistics assured the reliability of the transcriptome analysis and allowed further characterization of genes.

### 3.5. Differentially Expressed Genes in MBSes during Cold Storage

The expression abundance of each sample was measured, and differentially expressed genes (DEGs) were found between each set of two libraries. First, we normalized the read density measurement, and then used FDR (false discovery rates) < 0.01 and the absolute value of |log2 Fold Change| ≥ 1 as criteria to judge the statistical significance of gene expression. A large number of DEGs were obtained by comparing gene expressions between each set of two different libraries. We found that the number of downregulated DEGs was always slightly higher than upregulated DEGs in these three sets (Figure 4A). As shown in Figure 4B, compared to each set of two sample libraries, 5,508 (40.0%), 11,629 (76.7%), and 9,525 (62.8%) DEGs were predicted from “1_vs_3”, “1_vs_5”, and “3_vs_5”, respectively. Moreover, these DEGs were divided into four groups according to their different expression profiles. Group I was defined as upregulation, group IV as downregulation, and groups II and III as irregular expression patterns (Figure 4C). Moreover, enriched genetic annotation for DEGs was analyzed, and the Cluster of Orthologous Groups (COG), Gene Ontology (GO), Kyoto Encyclopedia of Genes and Genome (KEGG), Swiss-Prot, TrEMBL, NCBI nonredundant protein (Nr), and NCBI nucleotide sequence (Nt) databases were annotated to describe the functions and metabolism of the genes compared to the transcriptome database (*P* ≤ 0.05, hypergeometric test).

### 3.6. Gene Ontology Annotation

Expressed genes of MBSes were searched against the GO database to categorize standardized gene functions (Figure 5). About 31,987 genes were acquired and categorized into the three main categories, then summarized into 88 subcategories on the basis of GO classification, with the categories falling under cellular component (19,698, 61.6%), molecular function (19,698, 65.4%), and biological process (20,311, 63.5%). For the cellular component category, the most prominent categories were cell (GO: 0044464) and cell part (GO: 0005623), followed by organelle. For molecular functions, binding (GO: 0005488) and catalytic activity (GO: 0003824) were most highly represented. Within biological processes, metabolic process (GO: 0008152) and cellular process (GO: 0051179) were the most noticeable, demonstrating that these unigenes were involved in some mainly metabolic activities in MBSes (Appendix A).

### 3.7. Functional Genes Involved in Cellulose and Xylan Biosynthesis in MBSes during Cold Storage

The expression patterns of the genes encoding the key enzymes involved in cellulose and xylan biosynthesis are shown in Figure 6. Cellulose synthase (CesA) has a key influence on regulating cellulose biosynthesis, and the expression level of unigenes encoding CesA showed higher levels in stage 1 than in stage 3 and stage 5. A total of 30 unigenes were involved in xylan biosynthesis, which was the main hemicellulose. Among them were 1, 8, 8, 2, 4, and 7 unigenes encoding FRA8, GATL, IRX9, IRX14, IRX15, and IRX10, respectively (Figure 6). We found that the expression levels of *PeUXS*, *PeIRX9*, and *PeIRX10* presented higher in stage 1 and stage 5 than in stage 3, while the expression levels of *PeFRA8*, *PeIRX14*, and *PeIRX10* showed an increasing trend from stage 1 to stage 5 (Figure 6).

### 3.8. Functional Genes Involved in Lignin Biosynthesis in MBSes

Unigenes involved in the phenylpropanoid pathway as related to lignin biosynthesis are shown in Figure 7. Genes encoding PAL, caffeic acid 3-O-methyltransferase (COMT), and caffeoyl caffeoyl-CoA 3-O-methyltransferase (CCoAOMT) had higher expression levels than those encoding other genes. A total of 12, 4, 7, and 7 unigenes encoded PAL, CAD, COMT, and CCoAOMT, respectively. The expression levels of *PePAL* and *PeCCoAOMT* were observed to be higher in stage 3 or stage 5 than in stage 1 (Figure 7).

### 3.9. Functional Genes Involved in Candidate Transcription Factors in MBSes

Transcriptional regulation plays a crucial role in regulating secondary cell wall formation in plants, including the NAC and MYB family [33,34]. In our study, 1 and 2 unigenes encoding MYB46 and MYB63, respectively, were identified, and transcription analysis showed that the expression level of 2 unigenes encoding the MYB63 transcription factor showed a gradually higher gradient from stage 1 to stage 5 (Figure 8). Additionally, the expression level of 4 unigenes encoding NST1/2 transcription factor were observed to be decreasing in gradient from stage 1 to stage 5. Here, 4 and 2 unigenes encoding SND2 and KNAST7, respectively, showed the highest expression levels in stage 5 (Figure 8).

### 3.10. Expressional Analysis of Functional Genes Involved in Cell Wall Biosynthesis

A total of 15 candidate genes were identified among those involved in cell wall biosynthesis, of which 1 gene was involved in cellulose synthesis, 6 genes in lignin synthesis, and 7 genes in xylan synthesis (Figure 9). Expression analysis using qRT-PCR was also performed to compare the relative transcript levels of the unigenes in three stages of MBSes at 4 °C. *PH01000268G0210* encoding UXS and *PH01001724G0360* encoding C3H exhibited a higher expression level in stage 3 and stage 5 than those in stage 1 (Figure 9). *PH01000822G0290* encoding IRX9/IRX9-L was observed to be relatively high in stages 1 and 5 (Figure 9). *PH01000383G0390* encoding COMT, *PH01004840G0070* encoding CAD, and *PH01001342H0270* encoding MYB20 exhibited the highest expression levels in stage 3 (Figure 9).

## 4. Discussion

### 4.1. The Effects of Cold Storage on Cell Wall Composition in Post-Harvest MBSes

In this study, the longer we stored bamboo shoots at 4 °C, the higher the contents of cellulose and lignin that were found in MBSes were, while the content of hemicellulose became lower (Table 1). This indicated that cold storage slowed down, but could not stop, the xylogenesis process in MBSes. It is known that at room temperature, lignification in MBSes occurs rapidly [35]. Awano et al. [35] further showed that lignin deposited simultaneously with xylan penetration and microfibrils with globular xylan were masked by lignin, resulting in a homogeneous appearance of the cell wall. Song et al. [36] also found that heteroxylans notably increased in asparagus spears during cold storage. Lignin, involving the action of several primary phenylpropanoid metabolisms and lignin biosynthetic branching enzymes, is a complex polymer of phenylpropanoid mainly deposited in the cell wall [37]. The increasing lignin was processed by PAL, CAD, and POD, which brought about increasing enzymatic activities (Figure 2). Thus, the xylogenesis of MBSes correlated with the increase of lignin and cellulose contents and PAL, CAD, and POD activity. Interestingly, we detected xylan XylT activity in the middle of MBSes and found that xylan XylT activity decreased during the storage of 4 °C (Figure 3). This result was not consistent with our previous reports on asparagus, which found a high level of xylan XylT activity after harvest during storage [36].

### 4.2. Effects of Cold Storage on Molecular Mechanism of xylogenesis in MBSes

Transcriptome sequencing is a useful tool for rapidly obtaining information on the expressed fraction of a genome and is used to discover genes that control economically important traits [38,39]. By using the Illumina HiSeqTM 2500 sequencing platform and selecting the Moso bamboo genome database as a reference, we analyzed transcriptional changes related to secondary wall formation and the xylogenesis of MBSes during cold storage. A total of 12, 4, 7, and 7 unigenes encoding PAL, CAD, COMT, and CCoAOMT, respectively, were identified, and the expression levels of *PePAL* and *PeCCoAOMT* were observed to be increasing with the cold storage as it progressed (Figure 7), indicating key roles in lignin biosynthesis after harvest, as evidenced by high lignin contents and PAL activity after 16 days of storage (Table 1 and Figure 2). Zhang et al. [23] have also reported that lignin biosynthesis occurred in the mature basal segment in MBSes.

Xylan is the major hemicellulose in the secondary cell walls of eudicots and in the primary and secondary cell walls of grasses and cereals. Xylan synthesis in *Asparagus* spears post-harvest has been demonstrated, and xylan biosynthetic genes, such as *AoIRX9*, *AoIRX9-L*, *AoIRX10*, *AoIRX14_A*, and *AoIRX14_B*, had a relatively high expression level during cold storage [36]. Consistent with our previous report on *Asparagus* spears [36], 30 unigenes involved in the biosynthesis of xylan, including *PeUXS*, *PeIRX9 PeIRX10*, *PeFRA8*, *PeIRX14*, and *PeIRX15*, were identified (Figure 6). The expression level by RT-PCR further suggested that *PeUXS*, *PeIRX9*, and *PeIRX10* played key roles in the xylogenesis of MBSes after harvest (Figure 9), which has been demonstrated in the overexpression of *PeIRX10* in *Arabidopsis* [40].

Furthermore, unigenes encoding transcription factors, including NAC/SND and MYB, were found in MBSes during cold storage (Figure 8): 2 unigenes encoding MYB63 were identified and showed a gradually higher gradient from stage 1 to stage 5 (Figure 8). In *Arabidopsis*, the overexpression of MYB63 was found to induce ectopic deposition of lignin, but not cellulose and xylan, whereas the dominant repression resulted in a reduction in secondary wall thickening and lignin deposition [41]. Mitsuda et al. [42] have reported that NST1 and NST2 regulated secondary wall thickening in *Arabidopsis*. Additionally, subsequent works have shown that NST1 functions as a master switch of fiber cell differentiation in *Arabidopsis* [18,43]. However, our results showed that unigenes encoding NST1/2 had the highest expression level in stage 1 and the lowest expression level in stage 5. Thus, an MYB-based transcriptional regulatory system might have modified secondary cell wall biosynthesis and was involved in the regulation of lignin biosynthesis of MBSes during cold storage. Research work about the role of TFs in the xylogenesis of MBSes needs to be further elucidated.

## 5. Conclusions

The xylogenesis process in the cell walls of MBSes during cold storage was analyzed. As the storage time progressed, lignin and cellulose contents increased and the activities of PAL, CAD, and POD increased generally. Using transcriptomic analysis, we provided the expression pattern of genes encoding the biosynthesis of lignin, cellulose, xylan, and NAC-MYB-based transcription factors and suggested the role of these genes in the xylogenesis in MBSes during cold storage. Additionally, expressional analysis of candidate genes further yielded insight into the understanding of the molecular regulation mechanisms of xylogenesis in MBSes during cold storage.

## Figures and Tables

**Figure 1 polymers-11-00038-f001:**
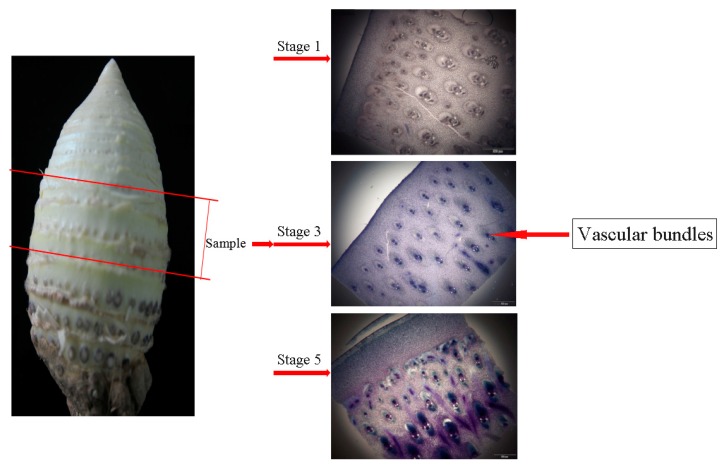
Transverse sections in middle sections of MBSes for three phases. The vascular bundles were stained blue. It showed no blue in color on day 0, whereas the vessels were stained from faint blue to oxford blue in the following two stages.

**Figure 2 polymers-11-00038-f002:**
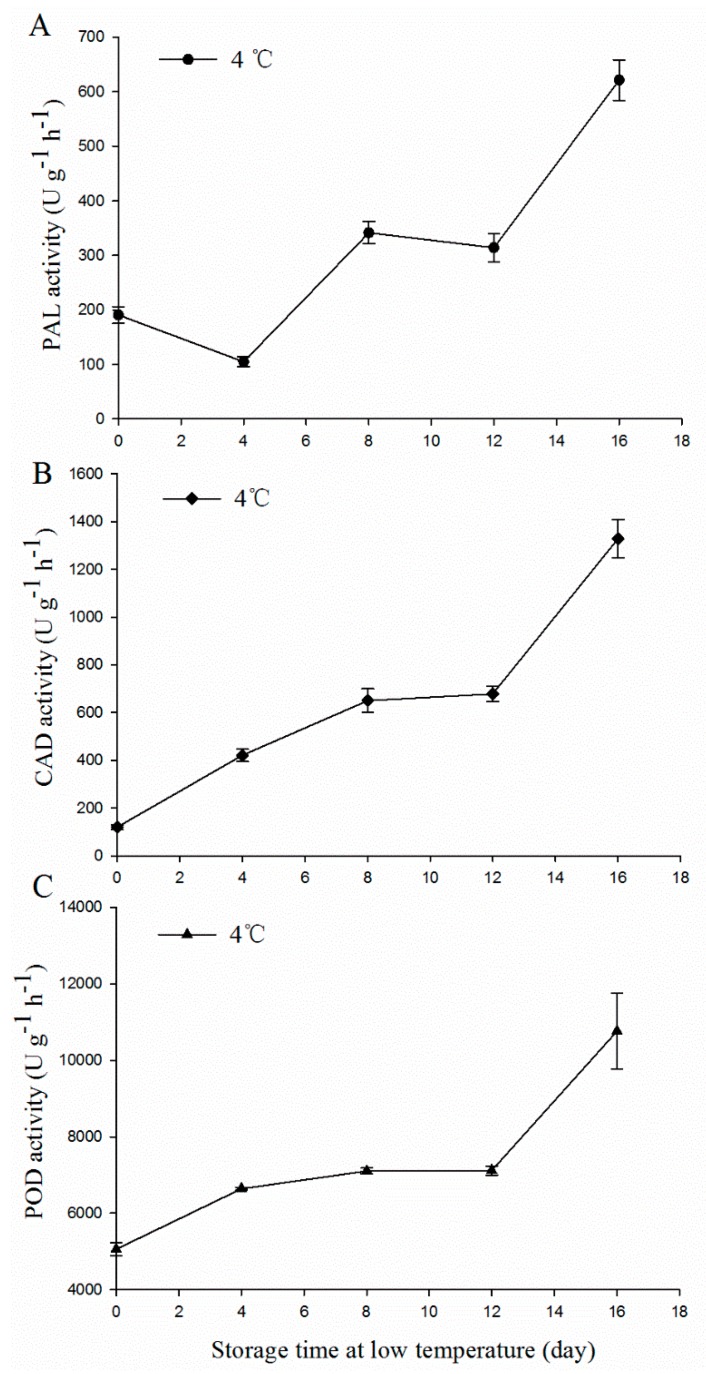
Activities of (**A**) phenylalanine ammonia lyase (PAL), (**B**) cinnamyl alcohol dehydrogenase (CAD), and(**C**) peroxidase (POD) in MBSes during cold storage. Data were average values ± standard error (SE) (*n* = 3).

**Figure 3 polymers-11-00038-f003:**
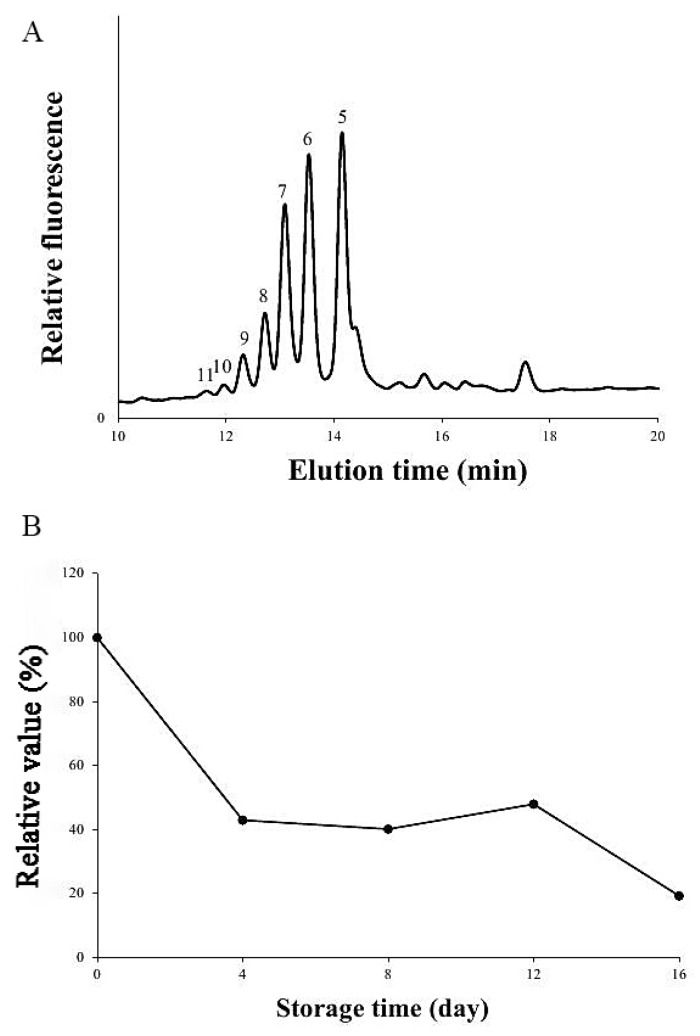
Xylan xylosyltransferase (XylT) activity (**A**) on day 0 measured using fluorescently tagged anthranilic acid (AA) Xyl_5_–AA as exogenous acceptors, and relative value, (**B**) compared to xylan XylT activity on day 0 of MBSes during the 4 °C storage period. The reaction was conducted by mixing MBS microsomes with cold UDP–Xyl and the fluorescent acceptors Xyl_5_–AA and incubated at room temperature (RT) for 6 h, and the reaction products were separated by reversed-phase high-performance liquid chromatography (RP)-HPLC and detected by a fluorescence detector. Microsomes from the middle of MBSes stored at 4 °C for 0–16 days were isolated, and the XylT activities were measured as described in “Materials and Methods”. The activity on 0 day is a reference. Data represent the mean of double measurements.

**Figure 4 polymers-11-00038-f004:**
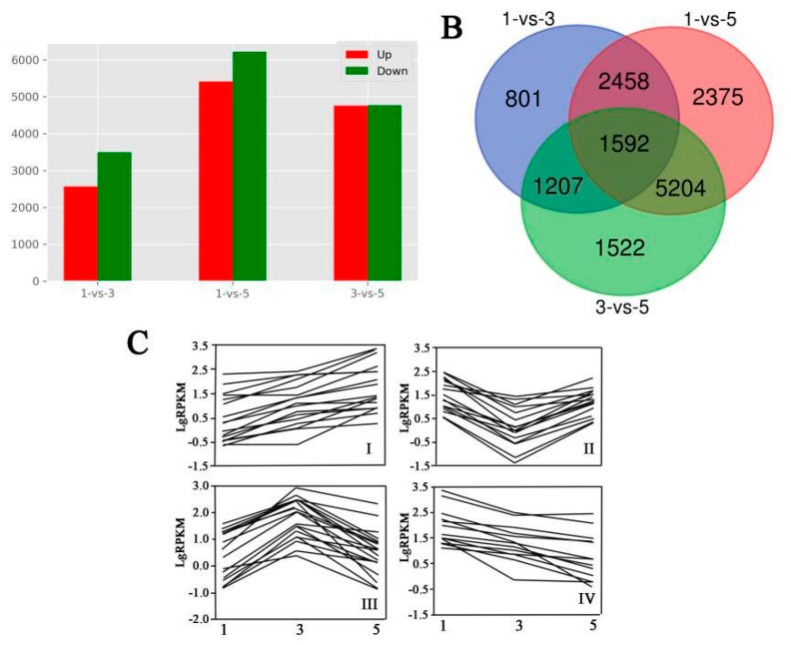
Analysis of mapped transcripts between two libraries. (**A**) Comparison of upregulated and downregulated differentially expressed genes (DEGs) in three sets. (**B**) Venn diagram representing the numbers of DEGs and the overlaps of sets obtained across three comparisons. (**C**) Changes in gene expression profile among the different segments.

**Figure 5 polymers-11-00038-f005:**
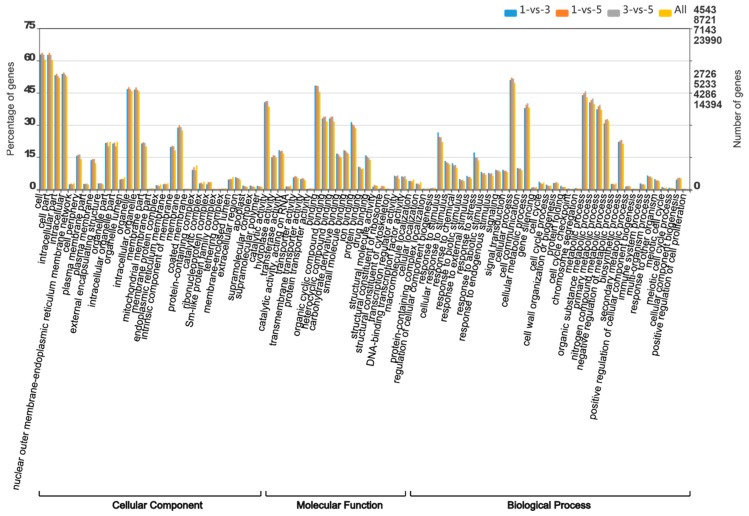
Three main categories in the Gene Ontology (GO) classification database.

**Figure 6 polymers-11-00038-f006:**
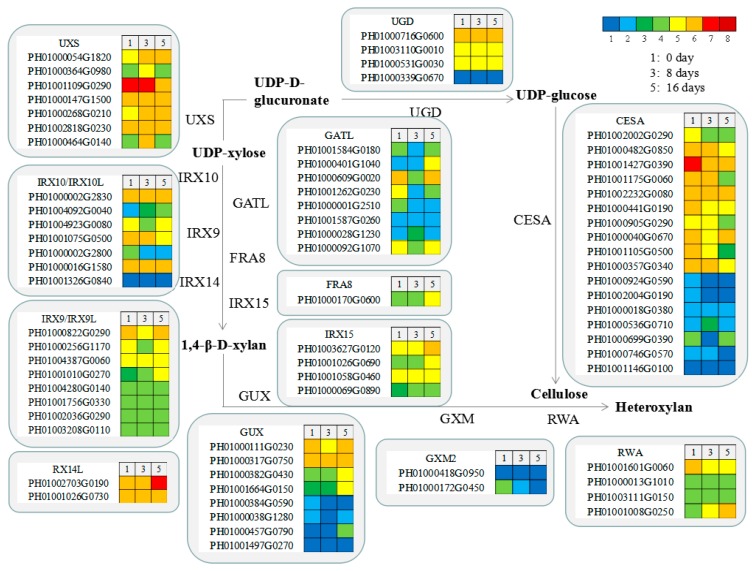
Genes involved in cellulose and xylan biosynthesis in different periods of cold storage of MBSes. Enzyme names, unigene IDs, and expression patterns are indicated in each step. The grids with eight different colors from blue to red show the reads per kilobase per million reads (RPKM) values. Here, 0-1, 1-3, 3-5, 5-10, 10-20, 20-30, 30-50, and over 50 are represented by colors 1 to 8, respectively.

**Figure 7 polymers-11-00038-f007:**
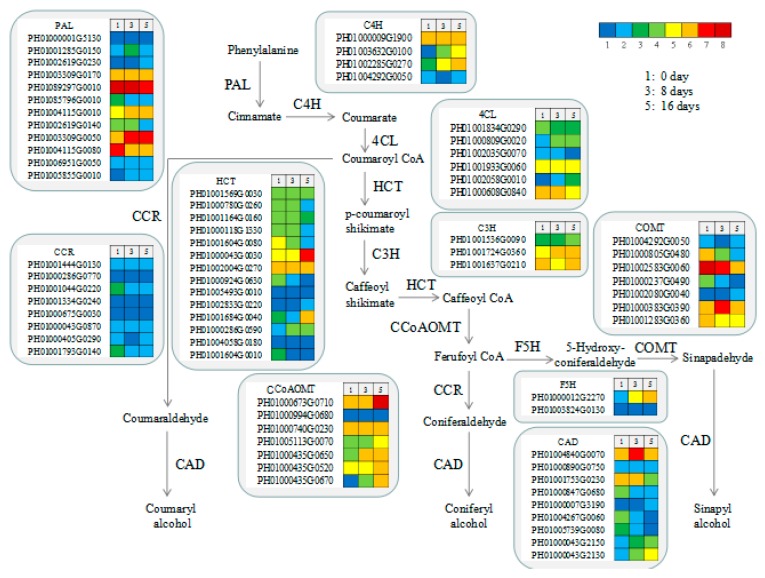
Genes involved in lignin biosynthesis in different periods of cold storage of MBSes. Enzyme names, unigene IDs, and expression patterns are indicated in each step. The grids with eight different colors from blue to red show the RPKM values. Here, 0-5, 5-10, 10-20, 20-40, 40-80, 80-160, 160-320, and over 320 are represented by colors 1 to 8, respectively.

**Figure 8 polymers-11-00038-f008:**
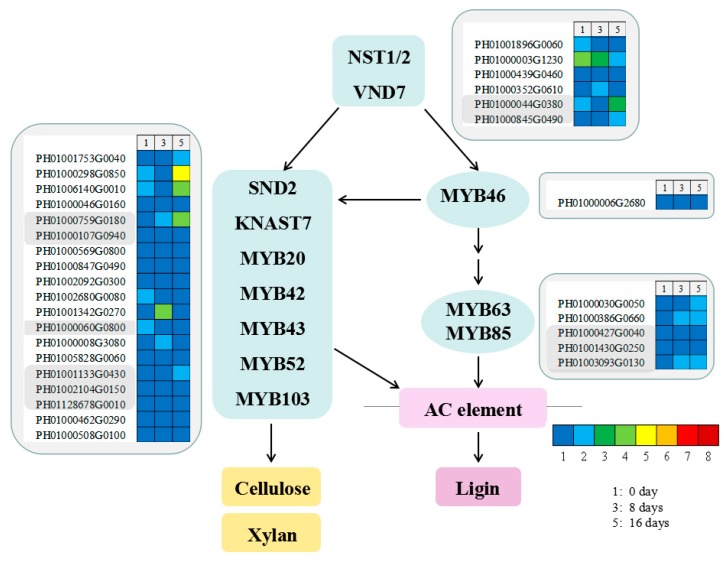
Unigenes in the transcriptional network regulating secondary cell wall biosynthesis and lignification according to *Arabidopsis thaliana*. Enzyme names, unigene IDs, and expression patterns are indicated in each step. The grids with eight different colors from blue to red show the RPKM values. Here, 0-0.01, 0.01-0.05, 0.1-0.5, 0.5-1, 1-3, 3-5, 10-15, and over 15 are represented by colors 1 to 8, respectively.

**Figure 9 polymers-11-00038-f009:**
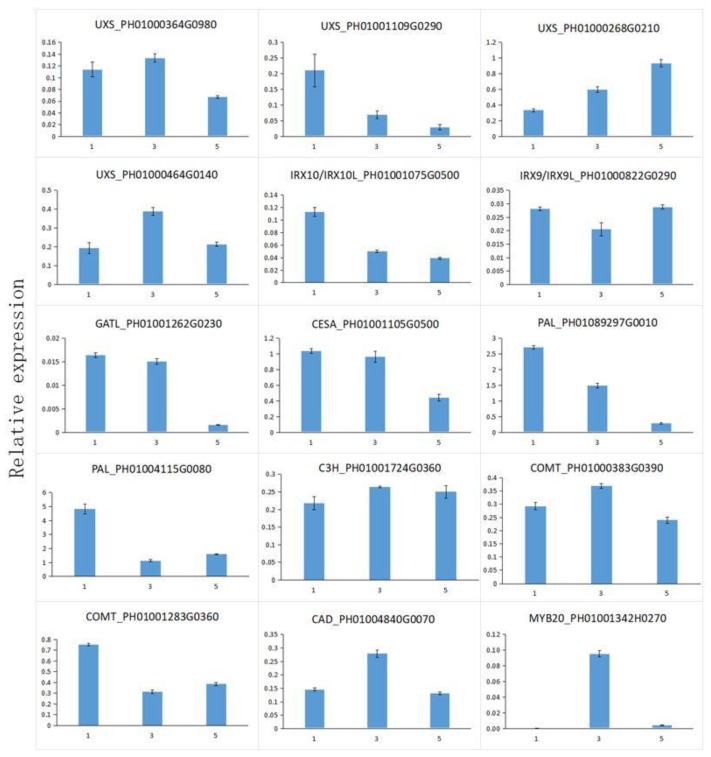
The expression profiles of the selected genes in MBSes during cold storage. The transcript levels were normalized to those of the nucleotide tract-binding protein (NTB), and the level of each gene in the control was set at 1.0. Each data represent the means ± SD (*n* = 3).

**Table 1 polymers-11-00038-t001:** Chemical composition of Moso bamboo shoots (MBSes) during storage at 4 °C. Data were average values ± SE (*n* = 3).

Days	Cellulose (%)	Hemicellulose (%)	Lignin (%)
0	24.10 ± 1.12	35.57 ± 0.09	11.85 ± 0.42
4	25.52 ± 1.00	33.76 ± 0.46	11.22 ± 0.26
8	26.24 ± 1.03	33.15 ± 1.12	13.70 ± 0.81
12	26.67 ± 1.04	31.67 ± 0.74	14.17 ± 0.34
16	28.90 ± 1.15	28.92 ± 0.61	15.62 ± 1.01

**Table 2 polymers-11-00038-t002:** Evaluation statistics of sequencing data in the three stages of 4 °C.

Sample	ReadSum	BaseSum	GC guanine and cytosine (G+C) content (%)	*Q*20 (%)	*Q*30 (%)
Stage 1	22,709,174	6,812,752,200	54.47	97.46	93.74
Stage 3	32,068,870	9,620,661,000	53.60	97.38	93.63
Stage 5	36,898,380	11,069,514,000	53.81	97.53	93.94

**Table 3 polymers-11-00038-t003:** Alignment statistics of MBSes for three phases.

Sample	Stage 1		Stage 3		Stage 5	
Statistical Content	Number	Percentage	Number	Percentage	Number	Percentage
Total reads	41,529,916	100.00%	44,048,144	100.00%	46,984,844	100.00%
Mapped reads	36,191,746	87.15%	38,481,918	87.36%	41,134,475	87.55%
Unique mapped reads	33,702,857	81.15%	36,257,017	82.31%	38,784,805	82.55%

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
