# Peer review of "Molecular Mechanism of Xylogenesis in Moso Bamboo (Phyllostachys edulis) Shoots during Cold Storage"

_polymers, 2018, doi:10.3390/polym11010038_

Round 1

Reviewer 1 Report

The authors studied the Molecular Mechanism of Lignification Related to Xylan XylT Activity in Moso Bamboo (Phyllostachys  edulis) Shoots during Cold Storage.

The paper is well written, and the experimental methods and results are clearly presented.

The work tries to investigate the MBS quality loss after harvest by insight into the mechanism of MBS lignification during cold storage process. Several methods and analysis have been applied to study the mechanism. Their in-depth analysis reveals important findings useful in harvest industry.

Overall, it is a nice contribution that meets the standards of Polymers and should be published after minor revision as indicated below:

-          Page 9, quality of Figure 3

-          Page 10, Figure 4, quality of part C

Author Response

We are grateful to the reviewing editor for his thoughtful and constructive comments. Modifications or clarifications have been made in accordance with comments and recommendations made by the reviewing editor. Major revisions in the manuscript are marked in red font.

Specific comments

Reviewer #1: minor revision

Comments 1: The authors studied the Molecular Mechanism of Lignification Related to Xylan XylT Activity in Moso Bamboo (Phyllostachys  edulis) Shoots during Cold Storage. The paper is well written, and the experimental methods and results are clearly presented. The work tries to investigate the MBS quality loss after harvest by insight into the mechanism of MBS lignification during cold storage process. Several methods and analysis have been applied to study the mechanism. Their in-depth analysis reveals important findings useful in harvest industry.

Response: Thanks for your comments.

Comments 2: Page 9, quality of Figure 3

Response: Thanks for your comments. As you suggested, we have improved the quality of the figures and please check them in the revised manuscript.

Comments 3: Page 10, Figure 4, quality of part C 

Response: Thanks for your comments. As you suggested, we have improved the quality of the figures and please check them in the revised manuscript.

Other revision has been also marked in red font in the revised manuscript. Also we have improved English language with the help of an English native speaker who has carefully edited the English language and scientific writing.

Reviewer 2 Report

This manuscript presented molecular mechanism study on the lignification process in Moso bamboo.  The experiment is well designed and the authors did lots of solid work. However, the data discussion and the structure of the content needs to be modified. Here are some suggestion to improve the draft.

Check through the draft to make sure the front and format are consistent. For example, lines 26&27 are different from the other content in the abstract.

Higher quality figures are required for Figures 3&4.

The data was simply described in section 3 and discussion was made in a separation section. It is very easy for a reader to lose the connection between the data and the discussion. 

The conclusion should be made in an independent section

Author Response

We are grateful to the reviewing editor for his thoughtful and constructive comments. Modifications or clarifications have been made in accordance with comments and recommendations made by the reviewing editor. Major revisions in the manuscript are marked in red font.

Reviewer #2:

Comments 1: This manuscript presented molecular mechanism study on the lignification process in Moso bamboo. The experiment is well designed and the authors did lots of solid work. However, the data discussion and the structure of the content needs to be modified.

Response: Thanks for your valuable comments. As you suggested, we have re-written the discussion by connecting the data with the discussion. Additionally, the structure of the content has been modified. Please check them in the revised manuscript.

Comments 2: Check through the draft to make sure the front and format are consistent. For example, lines 26&27 are different from the other content in the abstract.

Response: Thanks for your valuable comments. As your suggestion, we have double-checked the whole manuscript and corrected the front and format, including the lines 26&27 in the abstract section.

Comments 3: Higher quality figures are required for Figures 3&4.

Response: Thanks for your comments. As you suggested, we have improved the quality of the figures 3&4 and please check them in the revised manuscript.

Comments 4: The data was simply described in section 3 and discussion was made in a separation section. It is very easy for a reader to lose the connection between the data and the discussion. 

Response: Thanks for your comments. As your suggestion, we have re-written the discussion by connecting the data with the discussion. Please check them in the revised manuscript.

Comments 5: The conclusion should be made in an independent section  

Response: Thanks for your comments. As your suggestion, we separated the conclusion in an independent section. Please check them in the revised manuscript.

Other revision has been also marked in red font in the revised manuscript. Also we have improved English language with the help of an English native speaker who has carefully edited the English language and scientific writing.